# Transcranial Focal Electrical Stimulation Modifies Biogenic Amines’ Alterations Induced by 6-Hydroxydopamine in Rat Brain

**DOI:** 10.3390/ph14080706

**Published:** 2021-07-21

**Authors:** Cesar Emmanuel Santana-Gómez, Daniel Pérez-Pérez, Daniel Fonseca-Barriendos, Oscar Arias-Carrión, Walter Besio, Luisa Rocha

**Affiliations:** 1Department of Neurology, David Geffen School of Medicine at UCLA, Los Angeles, CA 90095, USA; csantanagomez@mednet.ucla.edu; 2Plan of Combined Studies in Medicine (PECEM), Faculty of Medicine, UNAM, México City 04510, Mexico; perez_dani18@hotmail.com; 3Pharmacobiology Department, Center for Research and Advanced Studies, México City 14330, Mexico; daniel.fonseca@cinvestav.mx; 4Unidad de Trastornos del Movimiento y Sueño, Hospital General Dr. Manuel Gea González, Secretaria de Salud, Mexico City 14080, Mexico; arias@iNeuron.mx; 5Department of Electrical, Computer, and Biomedical Engineering, University of Rhode Island, Kingston, RI 02881, USA

**Keywords:** non-invasive neuromodulation, 6-hidroxidopamine, dopamine, serotonin, histamine, transcranial focal stimulation

## Abstract

Transcranial focal stimulation (TFS) is a non-invasive neuromodulation strategy with neuroprotective effects. On the other hand, 6-hidroxidopamine (6-OHDA) induces neurodegeneration of the nigrostriatal system producing modifications in the dopaminergic, serotoninergic, and histaminergic systems. The present study was conducted to test whether repetitive application of TFS avoids the biogenic amines’ changes induced by the intrastriatal injection of 6-OHDA. Experiments were designed to determine the tissue content of dopamine, serotonin, and histamine in the brain of animals injected with 6-OHDA and then receiving daily TFS for 21 days. Tissue content of biogenic amines was evaluated in the cerebral cortex, hippocampus, amygdala, and striatum, ipsi- and contralateral to the side of 6-OHDA injection. Results obtained were compared to animals with 6-OHDA, TFS alone, and a Sham group. The present study revealed that TFS did not avoid the changes in the tissue content of dopamine in striatum. However, TFS was able to avoid several of the changes induced by 6-OHDA in the tissue content of dopamine, serotonin, and histamine in the different brain areas evaluated. Interestingly, TFS alone did not induce significant changes in the different brain areas evaluated. The present study showed that repetitive TFS avoids the biogenic amines’ changes induced by 6-OHDA. TFS can represent a new therapeutic strategy to avoid the neurotoxicity induced by 6-OHDA.

## 1. Introduction

The electrical modulation of the brain induced by strategies such as deep brain stimulation and vagal nerve stimulation has been considered an alternative treatment to control neurological disorders such as Parkinson´s disease and pharmacoresistant epilepsy [1,2,3]. However, these strategies of neuromodulation are invasive and have high economical costs [4,5].

On the one hand, transcranial focal stimulation (TFS) applied via tripolar concentric ring electrodes (TCREs) is a non-invasive strategy of neuromodulation that induces inhibitory and neuroprotective effects. TFS decreases the expression of the convulsive activity produced by pilocarpine, penicillin, and pentylenetetrazole [6,7,8,9]. TFS combined with sub-effective doses of diazepam reduces the incidence of mild and severe generalized pilocarpine-induced seizures, an effect associated with lower neuronal damage [10]. We recently found that TFS avoids the P-glycoprotein overexpression induced after an acute convulsive seizure and enhances the phenytoin effects in an experimental model of drug-resistant seizures [11]. The neuroprotective effects of TFS have been associated with lessened evoked glutamate release in vivo [12].

On the other hand, 6-hydroxydopamine (6-OHDA) is a toxin used to induce selective damage of catecholaminergic neurons and neurodegeneration of the nigrostriatal system [13]. The neurotoxicity induced by 6-OHDA is explained by the oxidative stress that results from the generation of reactive oxygen species (ROS) as a consequence of its auto-oxidation [14]. This condition associated with augmented free calcium can lead to cell death [15]. 6-OHDA is widely used to reproduce some of Parkinson’s disease signs as a result of neuroinflammation, neurodegeneration, apoptosis [16], and loss of dopaminergic neurons in the brain area injected [17].

In addition to the disruption of dopaminergic neurotransmission, 6-OHDA also modifies the neurotransmission mediated by other biogenic amines. The bilateral medial forebrain bundle 6-OHDA injection reduces serotonin tissue levels in the striatum [18]. 6-OHDA significantly augments histamine content in different brain areas (hypothalamus, hippocampus, and medulla oblongata) [19], a condition that facilitates neurodegeneration induced by the toxin [20].

It is described that neuromodulation strategies modify the neurotransmission mediated by histamine [21], serotonin, and dopamine [22]. The present study was designed to support the hypothesis that repetitive application of TFS can reduce the neurotoxic effects induced by 6-OHDA. Experiments were designed to evaluate the tissue content of dopamine, serotonin, and histamine in different brain areas of animals treated with an intrastriatal injection of 6-OHDA and subsequent repetitive application of TFS.

## 2. Results

### 2.1. TFS Did Not Alter the Tissue Content of Biogenic Amines

The Sham group showed the following tissue content values of biogenic amines expressed in µM/mg of protein, ipsi- and contralateral to the place of vehicle injection: a) dopamine: striatum, 186 ± 18 and 229 ± 20, respectively; cortex, 11.6 ± 1.8 and 24.2 ± 4.3; hippocampus, 12.5 ± 1 and 15 ± 1.7, respectively; amygdala, 13.9 ± 2.3 and 18.3 ± 1.2, respectively; b) serotonin: striatum, 57 ± 3 and 62.9 ± 6.2, respectively; cortex, 57.9 ± 7.5 and 68.6 ± 9; hippocampus, 56.6 ± 13 and 70.1 ± 6.1, respectively; amygdala, 157.4 ± 23 and 174 ± 10, respectively; c) histamine: striatum, 14.4 ± 1.6 and 13.6 ± 1.2, respectively; cortex, 11.8 ± 0.7 and 14.2 ± 1.6; hippocampus, 14 ± 1.1 and 14.2 ± 1.7, respectively; amygdala, 12.7 ± 0.7 and 14.3 ± 1.4, respectively. TFS did not induce behavioral changes during its repetitive application to the TFS group. The tissue content of dopamine, serotonin, and histamine in the different brain areas of the TFS group was similar to the Sham group (Figure 1, Figure 2 and Figure 3). These results suggest that TFS does not modify the normal brain function.

### 2.2. Intrastriatal Administration of 6-OHDA Modifies the Brain Tissue Content of Biogenic Amines

In contrast to the Sham group, animals from the 6-OHDA group showed a lower tissue content of dopamine in all the brain areas evaluated. This effect was more evident in the striatum, ipsi- and contralateral to the 6-OHDA administration (96% and 97%, respectively, *p* < 0.0001 vs Sham group) (Figure 1). Similarly, the tissue content of serotonin was lower in all cerebral structures analyzed (Figure 2). Concerning histamine and when compared with the Sham group, the 6-OHDA group presented high tissue content in ipsilateral striatum (80%, *p* < 0.0001), cortex (76%, *p* < 0.0006), and hippocampus (48%, *p* < 0.01); as well as bilateral amygdalae (ipsi- 82%, *p* < 0.0009; contra- 60%, *p* < 0.01). The experiments also revealed low histamine tissue content in the contralateral striatum (45%, *p* < 0.006) (Figure 3). According to the results obtained, 6-OHDA induces significant changes in the tissue content of biogenic amines in different brain areas, ipsi- and contralateral to a lesion.

### 2.3. TFS Modifies the Changes in the Tissue Content of Biogenic Amines Induced by the Intrastriatal Administration of 6-OHDA

Animals from the 6-OHDA-TFS group showed similarly low levels of dopamine tissue content in the striatum, ipsi- and contralateral to the side of toxin injection, when compared with the 6-OHDA group. However, the results obtained revealed that the other brain areas evaluated presented tissue content of dopamine similar to the Sham group. Indeed, the hippocampus and amygdala showed a significantly high tissue content of dopamine when compared with the 6-OHDA group (Figure 1).

The 6-OHDA-TFS group showed low serotonin tissue content in all the brain areas evaluated. However, when compared with the 6-OHDA group, higher values were detected in ipsi- and contralateral striatum (*p* < 0.02 and *p* < 0.05, respectively), as well as contralateral cortex (*p* < 0.001) and hippocampus (*p* < 0.001) (Figure 2).

Concerning histamine tissue content, animals from the 6-OHDA-TFS group presented values similar to the Sham group. When compared with the 6-OHDA group, lower values were detected in ipsilateral cortex, hippocampus, and amygdala (*p* < 0.03, *p* < 0.05, and *p* < 0.004, respectively) and contralateral amygdala (*p* < 0.02) (Figure 3).

The results obtained from the 6-OHDA-TFS group indicate that TFS applied daily for 2 h per day for 21 days does not avoid the loss of dopamine in the lesioned striatum. However, it protected peripheral structures from changes in biogenic amines induced by the toxin.

## 3. Discussion

The present study results indicate that the repetitive application of TFS for 21 days in normal animals did not modify the tissue content of dopamine, serotonin, and histamine in the different brain areas evaluated. However, repetitive TFS application avoids some of the long-term changes induced by the intrastriatal injection of 6-OHDA in biogenic amines in the striatum, cortex, hippocampus, and amygdala. 

It is known that the dopamine and serotonin tissue levels in the striatum and other brain areas such as the cerebral cortex and the hippocampus, decrease as a consequence of 6-OHDA intrastriatal injection [23,24,25]. 6-OHDA selectively destroys dopaminergic neurons due to its high affinity to the dopamine transporter [26,27]. 6-OHDA can also destroy adjacent cells as a result of the formation of ROS [16,27]. This situation explains the simultaneous changes in dopamine and serotonin in different brain areas induced by 6-OHDA. In contrast, we found that 6-OHDA induces enhanced tissue levels of histamine in different brain areas. This effect is similar to the results previously reported by other authors in rats [19,28] and patients with Parkinson’s Disease [29,30]. The high tissue content of histamine may contribute to the neuroinflammation process produced by the toxin administration [31].

Our results reveal that the repetitive application of TFS for 21 days avoided some of the changes of the biogenic amines tissue content induced by the striatal injection of 6-OHDA. These protective effects are similar to those induced by other strategies of neuromodulation. Deep brain stimulation improves the survival of dopaminergic neurons and avoids alterations in the tissue content of dopamine and serotonin in patients with Parkinson’s Disease [22]. Likewise, direct current stimulation applied for 4 weeks prevents the depletion of dopaminergic neurons induced by 6-OHDA, improving the motor alterations in the animals [32].

In the present study, animals receiving 6-OHDA and TFS showed a higher tissue content of dopamine in the different evaluated brain areas, when compared with the 6-OHDA group. Similarly, neuromodulation strategies such as high-frequency stimulation in 6-OHDA lesioned rats augment the extracellular levels of dopamine in the striatum [33]. 

It is known that behavioral and mood alterations in patients with Parkinson´s Disease are associated with reduced plasma levels of serotonin, suggesting its low neurotransmission [34]. Studies indicate that deep brain stimulation decreases serotoninergic neurotransmission [35,36,37]. Electrical stimulation of the subthalamic nucleus, a strategy used to reduce motor dysfunction in Parkinson´s disease patients, results in more evident behavioral and mood alterations [38,39]. This evidence supports that deep brain stimulation exacerbates the reduced serotoninergic neurotransmission associated with Parkinson´s Disease. In the present study, we found that rats from the 6-OHDA-TFS group showed higher tissue content of serotonin in contrast with the 6-OHDA group. Our results support that TFS is more effective than deep brain stimulation to avoid serotonin changes induced by 6-OHDA.

Studies indicate that high histamine levels in arkinsonian basal ganglia circuitry represent a compensatory mechanism for regulating the firing patterns of subthalamic neurons and decreasing motor dysfunction [40]. Nevertheless, the increased levels of histamine in other brain areas could lead to neuroinflammation with a consequent neuronal degeneration found in neurological disorders such as Parkinson´s disease [31]. Indeed, it is known that 90% of the histamine in brain areas such as the hippocampus is released from mast cells [41] that participate in neuroinflammation and cell damage [42,43]. In the present study, we found that TFS avoids the increased levels of histamine-induced by 6-OHDA in the ipsilateral striatum, cortex, and hippocampus and bilateral amygdalae. These results suggest that TFS represents a neuroprotective strategy in the 6-OHDA lesioned rats. 

It is also important to mention that the repetitive application of TFS in normal animals did not produce changes in the brain tissue content of the biogenic amines evaluated. In contrast, the repetitive application of transcranial magnetic stimulation increases the striatal dopamine levels in control rats [44]. This group of evidence supports that TFS represents a good neuromodulatory strategy to avoid the toxic effects of 6-OHDA in the brain. 

The unilateral application of 6-OHDA induces cell death and oxidative stress [45], both of these phenomena could be associated with changes in the biogenic amines. The present study presents important limitations for possible clinical implications of the TFS. One limitation is the lack of behavioral and histological evaluation. Future studies should be carried out to test whether TFS can avoid the behavioral and cellular changes induced by 6-OHDA. Another important limitation is that there was a limited number of animals used for each experimental group. Although this situation restricts the significance of the data obtained, the effects of TFS on the tissue levels of biogenic amines are evident. 

## 4. Materials and Methods

### 4.1. Animals

Adult male Wistar rats, initially weighing 250–300 g, were individually housed at 22 °C and maintained on a 12-h light/dark cycle. Animals had free access to food and water. According to the Mexican Official Standard (NOM-062-ZOO-1999) and the Ethical Committee of the Center for Research and Advanced Studies (Protocol #175/15 with approval date of 04/05/2016), the present study was carried out.

### 4.2. Surgery and Intrastriatal Injection of 6-OHDA

Rats were anesthetized with a mixture of ketamine (80 mg/kg, i.p.) and xylazine (20 mg/kg, i.m.). Through a trepan in the skull, an intrastriatal injection of 6-OHDA was applied (4 μg/μL in ascorbic acid 0.01%, 0.9% de NaCl, total 4 μL). The toxin was injected slowly (0.5 μL /min) into the right striatum (AP, +0.7 mm from Bregma; lateral 3.8 mm; ventral from the dura 5.0 mm). Then, a 6 mm diameter TCRE attached to male connector pins was placed centered on the cranium, 5 mm behind Bregma. Four stainless steel screws were threaded into the cranium over the frontal and cerebellar cortices and the electrode assembly was affixed to the skull with dental acrylic. Animals were allowed to recover for 2 days before any further manipulation.

### 4.3. Experimental Groups

**6-OHDA-TFS group (*n* = 6).** Two days after implanting and injection with 6-OHDA, animals received TFS daily for 2 h per day for 21 days. TFS consisted of 200 µs symmetrical biphasic square charge-balanced constant current pulses at a rate of 300 Hz and an intensity of 100 µA. TFS was applied through the outer ring (external diameter of 6.0 mm) and the disc of a TCRE (with the middle ring floating). For this purpose, we used two Grass Technologies S48 square pulse stimulators each with an SIU-C constant current stimulus isolation unit (Grass Technologies, West Warwick, RI). One S48 provided the positive pulses and the other provided the negative pulses. The day after the last TFS, rats were killed by decapitation, and the brain was obtained. Striatum, cortex, amygdala, and hippocampus, ipsi- and contralateral to the place of 6-OHDA injection, were dissected and immediately stored in polypropylene Eppendorf tubes at -70 °C.

**6-OHDA group (*n* = 6).** Rats were manipulated as indicated previously for the 6-OHDA-TFS group, except that they did not receive TFS.

**TFS group (*n* = 6).** Rats were manipulated as indicated previously for the 6-OHDA-TFS group, except that they received an intrastriatal injection of the vehicle instead of 6-OHDA.

**Sham group (*n* = 6).** Rats were manipulated as indicated previously for the 6-OHDA group, except that they received an intrastriatal injection of the vehicle instead of toxin.

### 4.4. Evaluation of Tissue Content of Biogenic Amines by Chromatography

Brain tissue samples (10–100 mg each) were homogenized on ice in 90 µL HClO4 0.1 M using a tissue homogenizer. The homogenate was centrifuged at 13,000 rpm for 15 min at 4 °C and the resulting supernatant was filtrated by a Nalgene-nylon 0.45 µm filter and stored at −70 °C until use. 

Then, 20 μL of the stored sample was injected into the solvent stream of a high-performance liquid chromatography system. Dopamine and serotonin levels were eluted using a reversed-phase column (C18, 3 μm; 2.1 × 50 mm; Atlantis, Waters^®^, Milford, MA, USA) coupled to a pre-column (Atlantis, Waters^®^, Milford, MA, USA) and mobile phase solution containing sodium acetate 25 mM, EDTA 0.01 mM; citric acid 25 mM and 1-octane sulfonic acid 1 mM dissolved in milli-Q water and mixed with acetonitrile in a proportion of 95:5, pH 3.35 ± 0.05 at a flow rate of 0.35 mL/min. Dopamine and serotonin detection was performed by a single-channel electrochemical detector (Waters^®^ model 2465, Milford, MA, USA) at 450 mV at 30 °C. 

For histamine levels, 15 μL of the sample was mixed with 10 μL OPA-NAC solution pH 9.3 ± 0.05 [46] in a vial. It was eluted using a reversed-phase column (C18, 4 μm; 3.9 × 150 mm; Nova-Pak, Waters^®^, Milford, MA, USA). A Ternary elution system was employed in a gradient flow with aqueous acetate phosphate buffer pH 5.05 ± 0.05 (eluent A), HPLC grade acetonitrile (eluent B), and milli-Q water (eluent C) [47]. The detection of histamine was performed by fluorescence detector (Waters^®^ model 474, Milford, MA, USA) with an excitation wavelength of 360 nm, an emission wavelength of 450 nm, and a gain value of 100. Peak height measurements quantified all the biogenic amines against standard solutions.

The pellet obtained from the homogenization of each brain sample was resuspended in fresh NaOH 0.1 N, and protein levels were determined by the method of Bradford [48]. The obtained values allowed expressing the data resulting from the high-performance liquid chromatography in µM/mg of proteins. 

### 4.5. Statistical Analysis and Sample Size

Previous studies indicate that six animals per group is the minimum sample size to detect significant differences in pilot studies evaluating quantitative variables [49]. According to this notion, the sample size for the present study was six rats per group.

Values were expressed as mean ± S.E.M. The results obtained were analyzed with ANOVA followed by the post hoc Tukey test. In all statistical comparisons, *p* < 0.05 was used as the criterion for significance.

## 5. Conclusions

TFS is a non-invasive neuromodulatory strategy with neuroprotective effects that also avoids most of the neurochemical abnormalities caused by the intrastriatal injection of 6-OHDA in the dopaminergic, serotoninergic, and histaminergic systems. Further research is needed to determine if TFS is able to protect the animals from the motor dysfunction and cell damage induced by the toxin.

## Figures and Tables

**Figure 1 pharmaceuticals-14-00706-f001:**
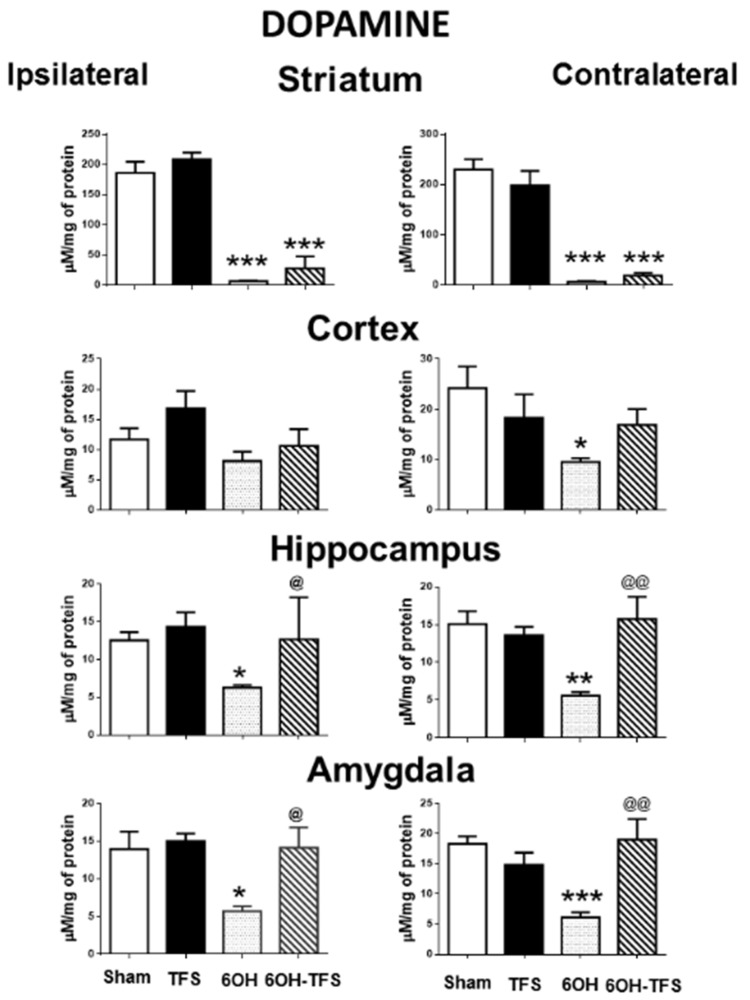
Effects of TFS on the changes in dopamine tissue content induced by 6-OHDA in different brain areas. Tissue content ipsi- and contralateral to the toxin injection or vehicle is expressed as µM/mg of protein. Sham, animals with vehicle injection; TFS, animals receiving TFS daily for 21 days; 6OH, rats injected with 6-OHDA; 6-OH-TFS, animals receiving repetitive TFS after 6-OHDA injection. Data are expressed as mean ± standard error. * *p* < 0.05 vs Sham group; ** *p* < 0.01 vs Sham group; *** *p* < 0.001 vs Sham group; ^@^ *p* < 0.05 vs 6-OHDA group; ^@@^ *p* < 0.01 vs 6-OHDA group. The results obtained were analyzed with ANOVA followed by the post hoc Tukey test.

**Figure 2 pharmaceuticals-14-00706-f002:**
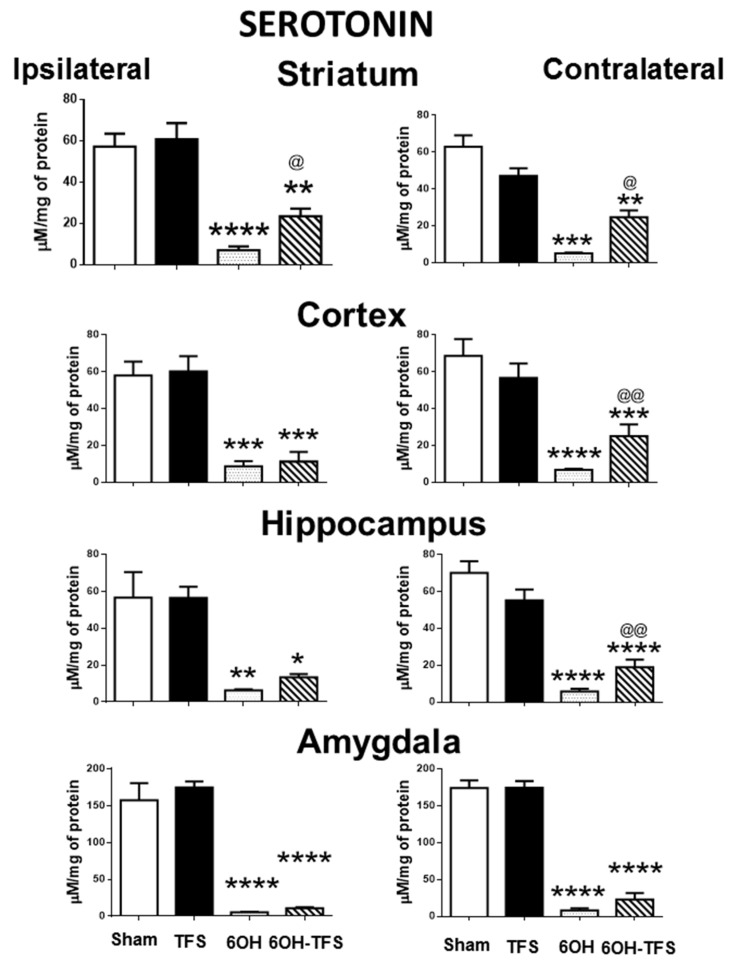
Effects of TFS on the changes in serotonin tissue content induced by 6-OHDA in different brain areas. Notations as in Figure 1. * *p* < 0.05 vs Sham group; ** *p* < 0.01 vs Sham group; *** *p* < 0.001 vs Sham group; **** *p* < 0.0001 vs Sham group; ^@^ *p* < 0.05 vs 6-OHDA group; ^@@^ *p* < 0.01 vs 6-OHDA group. The results obtained were analyzed with ANOVA followed by the post hoc Tukey test.

**Figure 3 pharmaceuticals-14-00706-f003:**
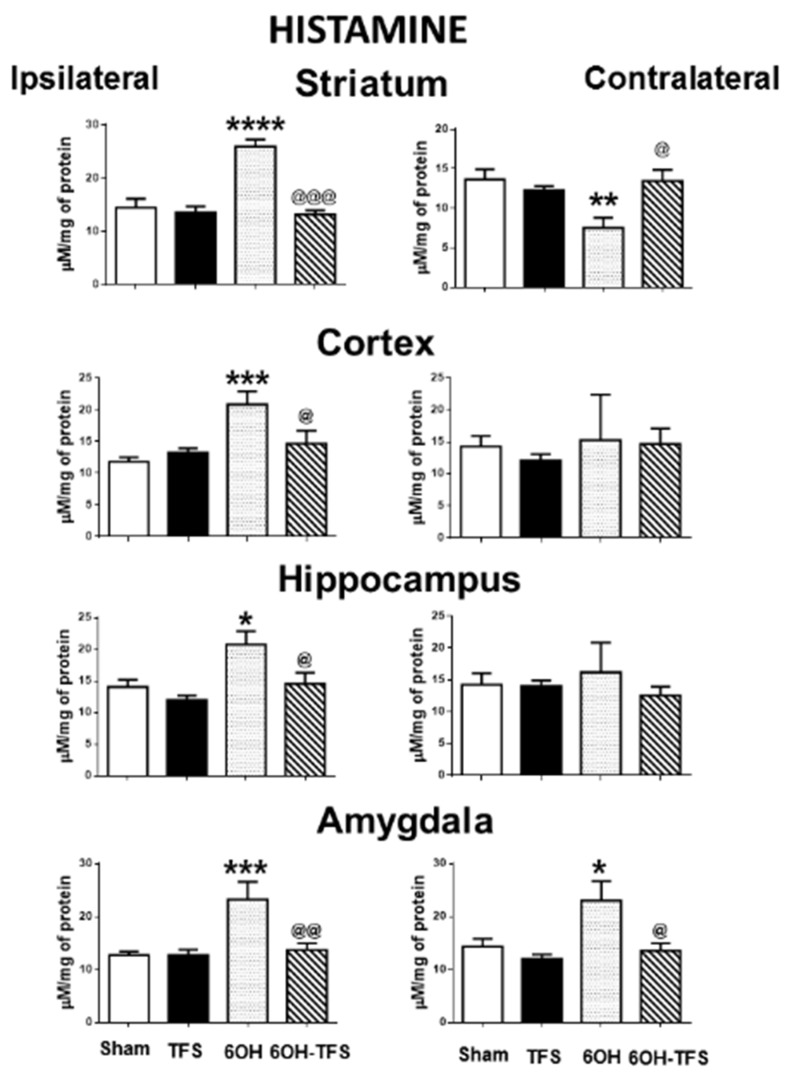
Effects of TFS on the changes of histamine tissue content induced by 6-OHDA in different brain areas. Notations as in Figure 1. * *p* < 0.05 vs Sham group; ** *p* < 0.01 vs Sham group; *** *p* < 0.001 vs Sham group; **** *p* < 0.0001 vs Sham group; ^@^ *p* < 0.05 vs 6-OHDA group; ^@@^ *p* < 0.01 vs 6-OHDA group; ^@@@^ *p* < 0.001 vs 6-OHDA group. The results obtained were analyzed with ANOVA followed by the post hoc Tukey test.

## Data Availability

Data is contained within the article.

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
