# Peer review of "Transcranial Focal Electrical Stimulation Modifies Biogenic Amines’ Alterations Induced by 6-Hydroxydopamine in Rat Brain"

_pharmaceuticals, 2021, doi:10.3390/ph14080706_

Round 1
Reviewer 1 Report
In this manuscript, authors reported protective effects of transcranial focal stimulation on damage induced by 6-hydroxydopamine intrastriatal administration as a model of Parkinson’s disease. This study was based on adult male rats killed 23 days after the 6-hydroxydopamine injection, following 3 weeks of transcranial stimulation. As only method, the investigators analyzed by liquid chromatography (HPLC) dopamine and serotonin (measured by an electrochemical detector), and histamine (using a fluorescence detector). These analytes were evaluated in the caudate-putamen, neocortex, hippocampus and amygdala. Dopamine levels were not affected by transcranial stimulation in the caudate-putamen ipsilateral to lesion, whereas serotonin was partially preserved and histamine was maintained at the basal values. Other effects were also reported for the contralateral side or in the other different cerebral regions. My comments are as following:
- This work does not present any evaluation of the structural damage; so, the reported effects could be interpreted as functional (i.e., they could depend on the different neurotransmitter release).
- Basically, there is no effect on the site directly damaged.
- No behavioral assessment of the lesion has been done (for instance, rotational behavior).
- The number of used animals is limited (n=6).
- As a minor issue, no mention of the statistical test used for the analysis is found in results and legends to the figures.
- Also, authors should state how animals were killed.
Author Response
Reviewer 1
In this manuscript, authors reported protective effects of transcranial focal stimulation on damage induced by 6-hydroxydopamine intrastriatal administration as a model of Parkinson’s disease. This study was based on adult male rats killed 23 days after the 6-hydroxydopamine injection, following 3 weeks of transcranial stimulation. As only method, the investigators analyzed by liquid chromatography (HPLC) dopamine and serotonin (measured by an electrochemical detector), and histamine (using a fluorescence detector). These analytes were evaluated in the caudate-putamen, neocortex, hippocampus and amygdala. Dopamine levels were not affected by transcranial stimulation in the caudate-putamen ipsilateral to lesion, whereas serotonin was partially preserved and histamine was maintained at the basal values. Other effects were also reported for the contralateral side or in the other different cerebral regions. My comments are as following:
Comment 1: This work does not present any evaluation of the structural damage; so, the reported effects could be interpreted as functional (i.e., they could depend on the different neurotransmitter release).
Response: We agree with the reviewer´s comment. The main goal of the present study was to demonstrate that TFS avoided several of the changes induced by 6-OHDA in the tissue content of dopamine, serotonin and histamine. We indicate that “Further research is needed to determine if TFS is able to protect the animals from the motor dysfunction and cell damage induced by the toxin”.
Comment 2: Basically, there is no effect on the site directly damaged.
Response: We agree with the reviewer´s comment that TFS did not avoid damage in striatum. However, through the discussion section it is described that the intrastriatal injection of 6-OHDA induces changes in adjacent brain areas such as cortex and hippocampus. Our results revealed that these changes were alleviated by TFS.
Comment 3: No behavioral assessment of the lesion has been done (for instance, rotational behavior).
Response: The main goal of the present study was to demonstrate that TFS avoided several of the changes induced by 6-OHDA in the tissue content of dopamine, serotonin and histamine. We indicate that “Further research is needed to determine if TFS is able to protect the animals from the motor dysfunction and cell damage induced by the toxin”.
Comment 4: The number of used animals is limited (n=6).
Response: We supported the number of animal used for the present study as follows: “Previous studies indicate that the minimum sample size to detect significant differences in pilot studies evaluating quantitative variables is 6 animals per group [49]. According to this notion, the sample size for the present study was 6 rats per group.” It is also indicated that this is a limitation of the study.
Comment 5: As a minor issue, no mention of the statistical test used for the analysis is found in results and legends to the figures.
Response: The following sentence is now included in the methods section and at the end of each figure legend: “The results obtained were analyzed with ANOVA followed by the post hoc Tukey test”.
Comment 6: Also, authors should state how animals were killed.
Response: In the section 4.3 Experimental groups, the description of the 6-OHDA-TFS group includes the following sentence: “The day after the last TFS, rats were killed by decapitation, and the brain was obtained”.
Reviewer 2 Report
I have read the manuscript entitled “Transcranial focal electrical stimulation modifies biogenic amines´ alterations induced by 6-hydroxydopamine in rat brain” (ID: pharmaceuticals-1234021) and I think that it is a valuable and interesting paper. The authors showed the effect of the repetitive application of transcranial focal electrical stimulation (TFS) on the dopamine, serotonin, and histamine changes in the selected brain structures, i.e., cerebral cortex, hippocampus, amygdala, and striatum, of mice treated with an intrastriatal injection of 6-hydroxydopamine (6-OHDA). They found that TFS does not affect biogenic amines concentrations in the analysed brain structures and could also reverse some 6-OHDA-induced changes in their levels which might be related to the neuroprotective effects. Further studies are planned to evaluate the influence of TFS on motor functions and cell damages in the 6-OHDA treated mice.
In my opinion, the study is well conducted and data are analysed and presented very appropriately. The manuscript is well written and does not need revision.
I think that the manuscript meets Pharmaceuticals' requirements and may be published in its present version
Author Response
Reviewer 2
I have read the manuscript entitled “Transcranial focal electrical stimulation modifies biogenic amines´ alterations induced by 6-hydroxydopamine in rat brain” (ID: pharmaceuticals-1234021) and I think that it is a valuable and interesting paper. The authors showed the effect of the repetitive application of transcranial focal electrical stimulation (TFS) on the dopamine, serotonin, and histamine changes in the selected brain structures, i.e., cerebral cortex, hippocampus, amygdala, and striatum, of mice treated with an intrastriatal injection of 6-hydroxydopamine (6-OHDA). They found that TFS does not affect biogenic amines concentrations in the analysed brain structures and could also reverse some 6-OHDA-induced changes in their levels which might be related to the neuroprotective effects. Further studies are planned to evaluate the influence of TFS on motor functions and cell damages in the 6-OHDA treated mice.
In my opinion, the study is well conducted and data are analyzed and presented very appropriately. The manuscript is well written and does not need revision.
I think that the manuscript meets Pharmaceuticals' requirements and may be published in its present version.
Response: We appreciate the reviewer´s comments.
Reviewer 3 Report
- The exact content of protein in µM/mg (mean ± sem) should be written in the paragraph related to the results (paragraph 2.1).
- You could increase the dimension of figure 1, 2 and 3.
- “Transcranial focal electrical stimulation (TFS) applied via tripolar concentric ring electrodes (TCREs) is a non-invasive strategy of neuromodulation that induces inhibitory and neuroprotective effects.”
Did you assess some effect of TFS on the gliosis? For instance, you could assess the effect of TFS, by using the immunohistochemical staining. To this aim, glial fibrillary acid protein (GFAP) is commonly used to assess gliosis in different animal models (Vinet et al., 2018; Costa et al., 2020). If not possible, you should at least discuss this point in the discussion.
References:
Vinet J, Costa AM, Salinas-Navarro M, Leo G, Moons L, Arkens L, Biagini G (2018) A hydroxypyrone-based inhibitor of metalloproteinase-12 displays neuroprotective properties in both status epilepticus and optic nerve crush animal models. International Journal of Molecular Sciences. doi: 10.3390/ijms19082178
Costa AM, Lucchi C, Simonini C, Rosal Lustoza Í, Biagini G (2020) Status epilepticus dynamics predicts latency to spontaneous seizures in the kainic acid model. Cell Physiol Biochem. doi:10.33594/000000232
- The limited number of animals could be a limitation. Could you please discuss this possible limitation in the discussion?
Author Response
Reviewer 3
Comment 1: The exact content of protein in µM/mg (mean ± sem) should be written in the paragraph related to the results (paragraph 2.1).
Response: The exact content of the biogenic amines obtained from the tissue of the Sham group is indicated in µM/mg of protein (mean ± sem) in paragraph 2.1.
Comment 2: You could increase the dimension of figure 1, 2 and 3.
Response: The dimension of the figures 1, 2 and 3 was augmented.
Comment 3: “Transcranial focal electrical stimulation (TFS) applied via tripolar concentric ring electrodes (TCREs) is a non-invasive strategy of neuromodulation that induces inhibitory and neuroprotective effects.” Did you assess some effect of TFS on the gliosis? For instance, you could assess the effect of TFS, by using the immunohistochemical staining. To this aim, glial fibrillary acid protein (GFAP) is commonly used to assess gliosis in different animal models (Vinet et al., 2018; Costa et al., 2020). If not possible, you should at least discuss this point in the discussion.
Response: We appreciate the reviewer´s comment. However, the role of gliosis in Parkinson's disease is contradictory. Indeed, it is described that patients with Parkinson´s disease show GFAP levels in the substantia nigra that correlate inversely with α-synuclein accumulation. According with these results, it is suggested that astrogliosis protects the neurodegeneration in Parkinson´s disease (Tong et al., 2015). On the other hand, ongoing experiments in our laboratory are focused to correlate the effects of TFS in neuroinflammation. Unfortunately, the results obtained are preliminary. We discussed the possible role of TFS in neuroinflammation according to the results obtained with histamine tissue levels obtained in the 6-OHDA-TFS group.
References: Tong J, Ang LC, Williams B, Furukawa Y, Fitzmaurice P, Guttman M, Boileau I, Hornykiewicz O, Kish SJ. Low levels of astroglial markers in Parkinson's disease: relationship to alpha-synuclein accumulation. Neurobiol Dis. 2015 Oct;82:243-253. doi: 10.1016/j.nbd.2015.06.010.
Comment 4: The limited number of animals could be a limitation. Could you please discuss this possible limitation in the discussion?
Response: We have indicated in the discussion section that the limited number of animals is a limitation of the present study.
Reviewer 4 Report
The paper entitled Transcranial focal electrical stimulation modifies biogenic 2 amines´ alterations induced by 6-hydroxydopamine in rat brain is well planned and organized. The abstract needs modifications with clear objective and and conclusions. In the introduction, please introduce hypothesis of the study. In result section, explain the result point wise and highlight significant findings The conclusion needs modification highlighting the major impact of the resultAuthor Response
Reviewer 4
The paper entitled Transcranial focal electrical stimulation modifies biogenic amines´ alterations induced by 6-hydroxydopamine in rat brain is well planned and organized.
Comment 1: The abstract needs modifications with clear objective and conclusions.
Response: Abstract was rewritten according to the reviewer´s comment.
Comment 2: In the introduction, please introduce hypothesis of the study.
Response: The last paragraph of the introduction section includes the following idea: “The present study was designed to support the hypothesis that repetitive application of TFS can reduce the neurotoxic effects induced by 6-OHDA.”
Comment 3: In result section, explain the result point wise and highlight significant findings
Response: The results are presented by section. Each section includes the results obtained from each experimental group. A conclusion is included at the end of each section.
Comment 4: The conclusion needs modification highlighting the major impact of the result
Response: Conclusion was rewritten according to the reviewer´s comment.
Round 2
Reviewer 1 Report
Authors did not perform the requested new experiments. Their findings are based on only one technique which presented a high variability. For instance, authors found more than 200% difference in neocortex by comparing the 2 hemispheres ("cortex, 11.6 ± 1.8 and 24.2 ± 4.3" in line 77), but this difference was not statistically significant. I still believe that this work requires a higher number of animals and more experiments.
Reviewer 3 Report
I don't have any other comments.